# Engaging Youth in Placemaking: Modified Behavior Mapping

**DOI:** 10.3390/ijerph17186527

**Published:** 2020-09-08

**Authors:** Sarah Little

**Affiliations:** Landscape Architecture Division, University of Oklahoma, Norman, OK 73069, USA; sarah.little@ou.edu

**Keywords:** participatory action research, PAR, children, behaviour mapping, behavioural mapping

## Abstract

Typically excluded from conversations about place, youth are becoming recognized as agents of change in placemaking. This article explores adapting a quantitative research method, behavior mapping, into a more youth-friendly qualitative participatory action research (PAR) method for placemaking projects, namely modified behavior mapping (MBM). The goal of MBM is to instigate placemaking conversations with youth with an understanding of an aspect of the lived experience of place and existing behavior. Sites are divided into observation zones, and youth are led through the zones by a trained facilitator. Like the quantitative method, MBM requires a list of behaviors of interest and a basemap. Behaviors are organized into groups on an observation sheet in a youth-friendly checklist format. A new checklist is printed for each observation zone. Basemaps can be an aerial photo or a downloaded map; however, creating a basemap by taking measurements will create science, technology, engineering, and math (STEM) learning opportunities. While in the observation zone, youth check the behaviors observed. Unlike the quantitative method, MBM does not require strict data collection protocols or a statistical analysis which makes the method more youth-friendly. Instead, MBM affords an opportunity for youth to reflect on their use of space and on others’ use of space. Results are disseminated through focus group discussions in order to create design programs or designs of place.

## 1. Introduction

Placemaking, “the participatory act of imagining and creating places with other people”, [1] transcends the act of beautification and addresses how a place can be transformed to reach its full potential to maximize use, create an identity, and positively alter the experience of place. The difference between “place” and “space” is important to note. Space is the dimensional structure of the physical environment while place represents the social and cultural meanings attached to space [2]. Place represents one’s experience of and meaning assigned to space. In other words, place represents the lived experience of space. The lived experience is defined by the “most mundane aspects of our daily lives” and not by epic historical events [3]. The lived experience of place manifests itself in everyday activities, such as behavior, routine, social interactions, and cultural rituals.

While the focus of placemaking activities is empowering people to imagine and create places, no particular focus as to the inclusive selection of the participants involved is specified in the definition. Certainly, there are examples of inclusive placemaking, but inclusion is not necessarily key to the process. Combining placemaking with Participation Action Research (PAR) methods expands the mission of placemaking to address diversity, equity, and inclusion. PAR seeks to generate action that will benefit a specific group of people which has been overlooked historically or does not possess the power or control for change within themselves [4]. To accomplish this, researchers utilizing PAR seek to guide participants in understanding their lived experience [4]. It is the lived experience that forms the bridge between placemaking and PAR. In the framework of placemaking, participants understand how to transform their experience of place by exploring their lived experience [5]. Again, the lived experience is captured in the everyday activities of life, such as behavior, routine, social interactions, and cultural rituals. Place is the stage on which these mundane activities occur. By transforming place through placemaking, everyday activities and the lived experience are transformed by extension.

More and more attention is focused on empowering youth to be agents of change in placemaking. Youth involvement in placemaking contributes to a sense of agency, to a feeling of acceptance within their communities, and to the creation of engaged citizens [6]. Placemaking initiatives that involve youth are becoming more commonplace, such as the Youth Voices program at the Aquinas Center in Philadelphia, PA [7], the Great Outdoors Colorado Inspire Initiative [8], and the design and renovation of the Manzanita Gathering Place in San Diego, CA [9]. All three examples involved placemaking and utilized PAR methods. Welch et al. (2020) describe a youth-led initiative in Philadelphia that utilized Photovoice and participatory mapping methods that resulted in youth leading members of the Philadelphia’s Planning Commission on a neighborhood tour where they identified threats and opportunities in an effort to influence policy [7]. Colbert (2020) details an initiative in Colorado where youth-led focus groups facilitated planning efforts to increase public access to nature [8]. Goldman (2020) recounts a project in San Diego where focus groups and art-based methods facilitated youth in designing and installing a park and creating hand-painted tiles for an art installation [9].

Many research methods fall within the purview of placemaking PAR with youth. In the Welch et al. [7], Colbert [8], and Goldman [9] examples, Photovoice, participatory mapping, focus groups, and art-based methods proved effective in working with youth ranging from 8 to 18 years old. Derr et al. (2018) identified photo documentation, art-based methods, such as drawings, murals, collage, and 3D models, interviews, focus groups, surveys and charrettes, short-term, intensive workshops where participants work with professionals to design a place, as suitable methods for children 2–18 years old [1]. Shamrova and Cummings (2017) conducted a literature review of PAR with youth and found that interview, focus groups, and photo elicitation were the most commonly used methods with children 10–18 years old [10].

While no dearth of methods exists for placemaking PAR with youth, the effectiveness of the method for placemaking is dependent on its ability to illuminate the lived experience. In order to understand how to transform the place in placemaking PAR activities, youth need to understand their lived experience of place. Focusing on behavior, an aspect of the lived experience, may provide a path to understanding. Behavior mapping is a quantitative method which associates behavior with place. The method involves the use of a precise map of the site and an observation sheet with a list of behaviors of interest. The researcher marks the location on the map of the observed behaviors with a number and records the behavior on the observation sheet with the associated number [11,12]. Ittelson, Rivlin, and Proshansky (1970) created the quantitative method, behavioral or behavior mapping, to observe how the physical environment of a psychiatric ward influenced behavior [13]. Results from the original study reported the physical environment in terms of the ward and room within the ward, e.g., bedrooms and public rooms; no basemap of the ward was provided. Behavior mapping evolved over time to include a basemap pinpointing the exact location of the behavior. The use of a basemap transformed behavior mapping in that patterns of behavior within the physical environment were highlighted during data analyses. Subsequent researchers have adapted the method to observe the behavior of children [11,12,14,15], wildlife [16], and the recovery of patients who suffered a stroke [17]. Planners, designers, and environmental educators, to name a few, utilize behavior mapping to document the usage patterns of a place, understand how usage changed after a renovation, or document the effectiveness of an outdoor educational program. Patterns of specific characteristics of place supporting specific behaviors become clearer with behavior mapping. Designers and planners can utilize knowledge gained from behavior mapping to design places which support desired behaviors. Refer to Cox et al. (2018) for a detailed description of behavior mapping [11].

The power of the quantitative method is the statistical association between the physical environment and behavior. Like any quantitative method, concerns of validity and reliability are present in behavior mapping. Researchers require extensive training in recording observations, and behavior mapping datasets are the result of many hours of labor-intensive data collection efforts since the more data collected translates to more statistical power [11]. While behavior mapping statistically elucidates the connection between place and behavior, the quantitative requirements of the method may be too great for young people. The demands of identifying statistical significance reduce its appropriateness as an activity for youth; however, the findings from behavior mapping would be valuable in placemaking PAR because of the connection between behavior and the lived experience.

In the article, a modification of behavior mapping is proposed in order to assist youth in placemaking PAR. Modified behavior mapping (MBM) transforms the quantitative method into a proposed PAR method. MBM, similar to behavior mapping, requires a list of behaviors to observe, a basemap, delineation of observation zones, opportunity to observe and collect data, and the need to analyze the data; however, the statistical demands are eliminated. In behavior mapping, the results are used to quantify the relationship between behavior and place. In MBM, the results are used to guide PAR activities within the framework of placemaking. For youth placemaking PAR projects, MBM becomes a tool that increases the awareness of behavior in the form of current site usage patterns, desired use of place as defined by youth, and the effect of their efforts, i.e., how behavior changed. MBM shifts the focus from quantifying the influence of place on behavior to exploring aspects of the lived experience of the place, regarding both current and desired behavior. Over the course of many years, the researcher has worked with youth in placemaking PAR activities such as leading charrettes where children designed a place, conducting focus groups to compile a lists of desired uses of a place, and capturing youth input about the current status of place during site visits. MBM was created from the knowledge gained from this experience. While the focus of the article is on youth, the method can be altered to accommodate a range of ages, from children to adults.

## 2. Materials and Methods

MBM requires the creation of an observation sheet of behaviors of interest, a basemap of the site, and a protocol for site observation. Identifying the behaviors to observe can happen through focus group discussions with youth, youth-conducted casual site observations, or can be provided by MBM facilitators. In focus groups, facilitators prompt youth to identify their current behavior at the place, e.g., “What do you like to do there?”. Additionally, facilitators prompt youth to identify their behavior when they were younger, e.g., “What did you like to do then?” The subsequent discussion focuses on the common behaviors across age groups, i.e., settings within the place that accommodate children of all ages. Additionally, youth can conduct a reconnaissance visit to the site and casually observe existing behaviors to compile the list. Regardless of the method of creating the list, blank spaces should be provided in the observation sheet to capture any unanticipated behavior.

Once the list of behaviors is finalized, the final observation sheet should be youth-friendly. One consideration is paper size. For ease of use, the observation sheet should be printed on standard sized paper that would fit on a standard sized clipboard. In the US, 8.5 × 11 inches is the standard letter size. Another consideration is the format of the observation sheet. A checklist format is a child-friendly option, i.e., the child would simply check off the behavior observed. One observation checklist is provided per observation zone, and the zone number is preprinted on the observation sheet (see Figure 1).

One potential problem could be a long list of behaviors may be visually overwhelming. Grouping behaviors into categories, e.g., play type (e.g., [18]) or physical activity level (e.g., [19]), could simplify the final checklist format. The behavior categories become steps on the checklist. In the example provided in Figure 1, behavior categories include physical activities (step 1 in Figure 1), e.g., running, jumping, and hanging; loose parts play (step 2 in Figure 1), e.g., playing with water, wood chips, and shovels; social activities (step 3 in Figure 1), e.g., talking, fighting, or playing alone. Going through the checklist step by step will ensure a consistent level of information is collected across the site. In addition to behavior, a rough estimation of how many people were visiting the zone and their approximate age should be included in the checklist. With this information, youth will understand how the place is used, e.g., which zones support specific behaviors, the age of the visitors in the zones, and which zones receive the most use.

A basemap in behavior mapping must be precise so that researchers can accurately record the location of the observed behavior in order to associate place characteristics with the behavior. In MBM, the exact location of the behavior is not as important since the general location of the behavior is of interest. For example, knowing that the play equipment, in general, supports climbing is more valuable in MBM than knowing the specific component of the play equipment that supports climbing. Basemaps in MBM can be aerial photographs, maps downloaded from the internet, or site maps provided by the managing body, such as a map of a playground provided by a Parks and Recreation Department or a map of an exhibit provided by a Zoo or Nature Center. Typically, only facilitators utilize the basemap during MBM observations. Primarily, basemaps are used to denote the observation zones within the site, so facilitators will know the boundaries of the zones and the order of rotation in order to lead youth through the exercise.

In some instances, a precise basemap may be necessary in placemaking PAR activities. Engaging youth in generating a basemap presents an excellent opportunity to become better acquainted with the site by cataloging existing conditions. Additionally, creating a basemap presents an opportunity to incorporate STEM (science, technology, engineering, and math) learning activities into placemaking PAR for older children. Drawing a basemap can teach youth STEM principles such as scales, cardinal direction and seasonal sun angles, plant identification, and arithmetic and geometry. Tapping into the STEM nature of placemaking may prove valuable in engaging STEM professionals in the project, accessing funding sources which emphasize STEM education, and connecting to STEM curriculum.

In general, making a basemap requires the establishing of site boundaries, graph paper large enough to accommodate a scaled drawing of the site, a clipboard that fits the dimensions of the paper, pencils with erasers, and measuring tapes of varying lengths. Regardless of the source of the basemap, the outermost boundaries of the site being observed need to be established. Boundaries of sites that contain edges, such as fencing, pathways, or trees, are easy to determine. For sites without these edges, a consensus among the placemaking partners should be reached regarding the boundaries of the site.

By understanding the general site dimensions, facilitators can select the appropriate size of graph paper. If a site is roughly 30′ × 40′, then an 8.5″ × 11″ graph paper at ¼ scale would suffice. A ¼ scale means that ¼″ on paper equals 1′ in reality; therefore, an inch on paper equals 4′ on the ground. An 8.5″ × 11″ sheet of graph paper at a ¼ scale would accommodate a site that is 34′ × 44′. Graph paper is available in many increments; however, increments based on the accepted unit of measurement will allow for the use of measuring tapes to measure drawings. For example, objects drawn on graph paper based on an increment of 10 would be difficult to measure with an imperial unit measuring tape.

Measuring the site to create the basemap requires measuring tapes. Shorter handheld tapes usually measure between 25′ and 35′ and longer reel tapes measure between 100′ to 300′. At least one short and one long measuring tape is recommended. The length of the tapes depends on the size of the site; larger sites would require longer tapes. When measuring the site, tapes, regardless of size, should be kept perpendicular to mimic the grid on the graph paper. For example, a 30′ × 40′ site is adjacent to a building. The longer tape is set at the corner of the building. Using the wall of the building as a sight guide, the longer tape is pulled along the length of the site staying in line with the building wall. Utilizing the shorter tape, elements within the site are located by measuring perpendicularly to the longer tape (see Figure 2). The best way to record the measurements is to draw the site to scale on the graph paper as the measurements are taken. Recording the measurements by simply writing the number tends to lead to confusion later when translating the written measurements into a drawing.

Once the outermost boundary of the site being observed has been established and a basemap is procured, the site is divided into smaller zones to facilitate observing behavior. Zones are determined based on many factors. Since facilitators guide youth through observation rotations, the number of facilitators available may influence the number of zones. High numbers of youth participating in placemaking activities may necessitate a higher number of zones. Youth should be able to stand in one location and observe the entire zone; sometimes, playground equipment is divided into multiple zones based on visibility. No area of the site within the established boundary should be outside of a zone. Natural edges such as borders, buildings, pathways, and trees act as boundaries of each zone [11]. For example, the safety surface surrounding play equipment could differentiate a zone.

The protocol for MBM is greatly simplified when compared to behavior mapping. Protocols refer to how the data are collected in order to ensure that the research is systematic and can be repeated by other researchers. Some protocol decisions in MBM include the order of zone rotation, length of observation time in each zone, and observing and data recording procedures. Zones are assigned an ordinal number to represent the order of rotation and should only be observed once per round. For example, a group that starts in zone 3 will rotate to zone 4 once the observation is completed. The observation round is completed once the group observes all the zones and stops after observing zone 2. Youth simply visit each observation zone, orderly, and check off behaviors they observed without tallying the instances of the behavior. Utilizing the example provided in Figure 1, youth go step by step through the observation sheet checklist, making sure every “step” has been completed. The zone number is specified on the observation sheet, and a new sheet is used in each zone. For ease of data collection and analysis, each checklist is prelabeled with a zone number. All observations for zone one are recorded on the observation sheet checklist prelabeled “zone 1”. At the beginning of MBM, each youth should be provided with a clipboard with the numbered observation sheets organized in order of rotation. In the previous example, the top sheet on the clipboard is the observation sheet for zone 3, the next sheet is zone 4, and the last sheet is zone 2.

Youth are led through the zone rotation by the trained facilitator. Once a full rotation or round is completed, the observation ceases or youth can go through the zones again. The purpose of MBM is not to generate many data points for statistical analysis but to capture the behavior supported by the place; therefore, multiple rotations may not be necessary.

The time spent observing each zone does not have to be equal. Zones that are heavily populated will require more time to observe than zones with no one. Observing zones with no or few people for long periods of time may dampen the momentum built from observing zones with many people. Even five minutes observing can seem an eternity to youth if no one is in the zone. Since time spent in each zone may vary, facilitators must ensure that only one group observes the zone at a time; therefore, the protocol must address the possible difference of times spent in the zones. Dividing the zones based on use may be a good option; high use areas may be divided into several zones while adjacent low use areas may be lumped together. Once the zone visits are completed, facilitators collect observation sheets and compile findings and promptly report to youth to maintain enthusiasm and interest.

Both methods benefit from trained personnel. As mentioned earlier, researchers are trained in collecting data in behavior mapping. Incorporating trained facilitators in MBM to assist youth during observations makes the process more efficient and effective. The facilitator guides focus group discussions within MBM, helps create a basemap, organizes the observation sheet on the clipboards, indicates observation zone boundaries to youth, ensures that observations for each zone are recorded accurately on the sheet labeled for that zone, guides youth through the zone rotation, and answers questions that may arise. Trained facilitators can be older youth involved in the project or trusted adults who have a rapport with the participants.

When utilizing MBM, some considerations include an ethics board review of the method and adjusting the method based on the age of the participants. Research conducted by academics that involves human subjects must be reviewed and approved by an ethics board in order to be published. In the US, universities assemble accomplished academics to form an Institutional Review Board (IRB) that reviews and approves methods involving human subjects. In the US, the IRB mandate is outlined in *The Belmont Report* compiled by the US Department of Health, Education, and Welfare [20]. Behavior mapping is typically exempt from IRB reviews since no identifiable information is collected and no interaction occurs between the researcher and person being observed. However, MBM will require an IRB review since youth are collecting the data and participating in focus group discussions. Typically, an IRB review requires that all personnel, e.g., MBM facilitators, involved receive IRB training. Beyond the methods receiving IRB approval, parental permission will likely be necessary. Additionally, if youth are under 12 years old, a signed assent form is typically required. Getting all facilitators trained, permissions signed, and the IRB review completed may add weeks or months to the project timeline.

The age of participants is another important consideration. The method can be adjusted to accommodate a range of participant ages. The presence of a trained facilitator may make the method appropriate for younger children. Based on previous experience with U.S. fifth graders (10–11 years old), the method would be appropriate as long as the facilitator had a good rapport with the children and the group size of observers was limited to no more than 2–3 children.

While quantitative and qualitative differences exist between all the steps of behavior mapping and MBM, the findings represent the main point of deviation between the two methods. In behavior mapping, the findings quantify the influence of place on behavior and vice versa. Behavior mapping findings inform future designs by identifying the behavior supported by specific characteristics of place, evaluating the success of design installations, e.g., post occupancy evaluations, and documenting the change in behavior before and after a renovation. In MBM, the findings also address the influence of place on behavior, however, the findings are utilized within a placemaking PAR framework to empower vulnerable people to transform their experience of place. Data collected from MBM are presented to youth in a focus group format where youth identify findings and determine how the findings influence the placemaking process. Once place has been transformed through placemaking, MBM can be utilized again to document the change in behavior and allow youth to recognize the influence of their participation on the placemaking process.

Focus groups are an effective forum for youth to identify the findings from MBM. Facilitators start the discussion with initial questions such as the following:Was the behavior expected? Did you observe anything unusual?Was the behavior typical of your experience at the place? Did you participate in similar behavior?Was the observed behavior consistent with the type of behavior you wish would occur at the place?

While facilitators may guide the discussion, youth should identify the meaningful findings from the data collected. Where the focus group discussion veers from here depends on the objective of the placemaking PAR project and the MBM findings. MBM is most useful to placemaking PAR projects that involve creating a design program or the design or renovation of a specific place through the charrette process.

Design programs represent a wish list of elements, activities, and uses of place; they account for “the quantity and quality of spaces needed to meet anticipated future needs” [21]. An effective design program assists in design development, whether the design is developed by youth in placemaking PAR activities or practitioners. MBM findings illuminate the connection between the characteristics of place and current behavior. During focus group discussions, youth develop design programs with MBM findings that identify the characteristics of place to preserve the desired behavior or opportunities to enhance an existing place to encourage a specific behavior.

A charrette is a short-term, intensive design workshop. Typically, a charrette is confined to a work week, a long weekend, or an afternoon. During that time, completing the design is the sole purpose. In the context of MBM and placemaking PAR, youth can create designs for new places or renovations to existing places. Charrettes involving the design of a new place may not benefit from MBM since there may not be any behavior to observe in an undeveloped site. Instead, youth can observe behavior at similar places. For example, if youth are involved in the design of a new park on undeveloped land, then they can observe behavior at existing parks and use those findings to inform the design process.

Charrettes that focus on renovating existing places benefit from the insight gained from MBM. Beyond the benefit of design program development and design creation, MBM can be performed after the renovation is completed in order for youth to understand the effect of their efforts. Through MBM, youth can document how the behavior changed pre- and post- renovation.

Creating a design program or a design/renovation based on current behavior alone may hinder the opportunity to create more inclusive places. During data collection, youth collect data about the approximate number of people within a zone and their approximate age group. An MBM finding could identify the absence of a certain age group. For example, youth did not observe teenagers at the place. Placemaking PAR participants can recommend that either practitioners, MBM facilitators, or themselves interview teenagers to ascertain characteristics that would attract them to the place. Identifying and interviewing nonusers through MBM can help to create more inclusive places.

Once the place has been transformed through placemaking, youth can conduct another round of MBM to document the change in behavior and subsequently their impact on the placemaking process. Again, youth record behavior and identify findings through focus groups. At this point, the findings do not inform the placemaking process which has concluded; instead the findings are more reflective in nature, focusing on their influence on placemaking. Facilitators ask questions which foster reflection about their role in the transformation, such as:Did the behavior change in expected ways?Did you observe any unexpected behavior?How did your participation influence the placemaking process?How can the placemaking process/MBM be improved?

The focus group discussions empower youth by illuminating the environmental and behavioral changes enacted by their involvement. The postplacemaking MBM exercise builds agency in youth participants when they realize that decisionmakers actively sought their input, their participation transcended tokenism in that their input informed the process, and their efforts transformed the place.

## 3. Conclusions

More and more placemaking PAR initiatives involve youth as agents of change. Empowering youth in the cocreated imagining of place fosters agency, feelings of acceptance within their communities, and a reality of engaged citizens for the future and provides a more comprehensive perspective of placemaking by giving a voice to this historically underrepresented group. Youth engagement in placemaking ensures the creation of more inclusive places.

While existing methods within PAR and placemaking facilitate youth participation, MBM differs in that youth assume a more active role in the process. Youth become the research instrument in that they collect the data and identify the findings. Through focus groups, youth utilize the findings to create recommendations that influence the placemaking process and result in the transformation of place. MBM engages youth through meaningful observations, discussions, and reflections which builds agency, connection to community, and engaged citizens in addition to building more inclusive environments.

Limitations of MBM include ethics board reviews, the reliance on facilitators, working within the time constraints of youth schedules, and ensuring a high degree of institutional support. As mentioned earlier, MBM will require review from an ethics board in order to publish results. Any research with vulnerable populations, such as children, receives extra scrutiny from ethics boards which can delay the project. Additionally, obtaining the required signed permissions from guardians and youth may delay the project. Finding and training facilitators may prove difficult since facilitators need to have an existing rapport with the youth involved in MBM. Today’s youth are busy; schedules may not allow full participation. Finding enough youth available to conduct MBM may be a challenge. Lastly, like any placemaking effort, success is incumbent on engaging decisionmakers. For the vision identified by placemaking PAR and MBM to be realized, successful efforts occur simultaneously from bottom-up and top-down efforts [1,22,23]. Bottom-up occurs when youth are motivated and empowered to enact change through grass-roots efforts; top-down occurs when the decisionmakers, the people in power, are engaged in the project and committed to implement youth-led change.

MBM affords an opportunity for youth to explore an aspect of their lived experience of place, i.e., behavior, and transform that experience. By simply creating an observation sheet checklist and a simplified basemap and recruiting and training facilitators, MBM revolutionizes placemaking PAR by illuminating how the place is currently being used and documenting the behavior change of the placemaking effort once completed. MBM affords a framework for which youth evaluate place, their position within said place, and their power to change said place.

## Figures and Tables

**Figure 1 ijerph-17-06527-f001:**
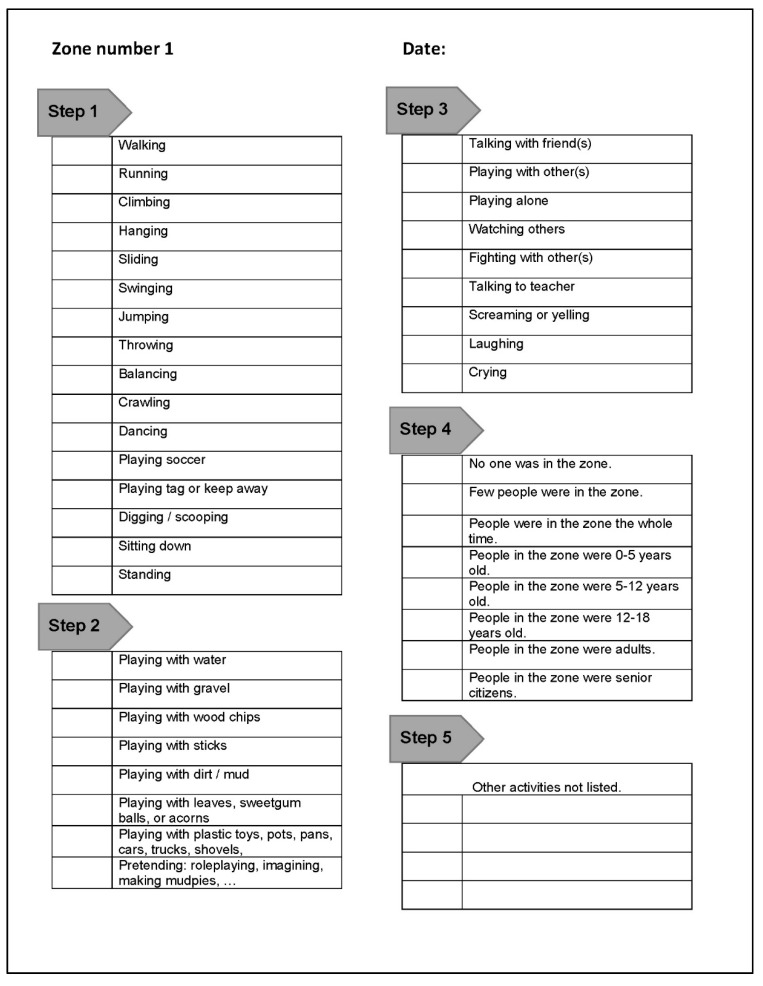
Utilizing a checklist as the format for the observation sheet ensures that the children collect a consistent level of information across the zones.

**Figure 2 ijerph-17-06527-f002:**
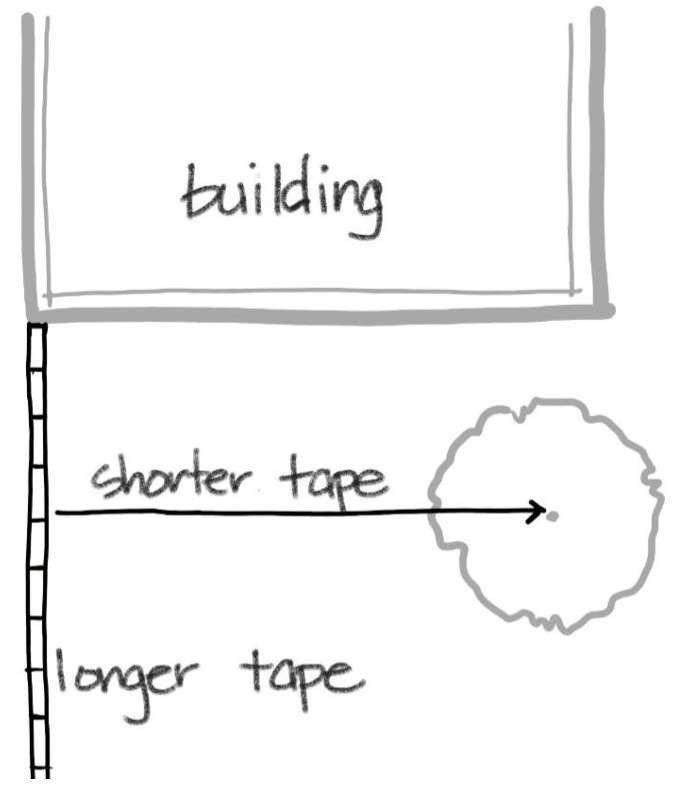
When creating a basemap, basic principles of geometry are useful. Shorter and longer measuring tapes should remain perpendicular to mimic the grid on the graph paper.

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
