# Peer review of "Engaging Youth in Placemaking: Modified Behavior Mapping"

_ijerph, 2020, doi:10.3390/ijerph17186527_

Round 1
Reviewer 1 Report
Overall, the paper is well written and addresses an important issue. The authors propose a qualitative method, namely modified behavior mapping (MBM) for place making projects. However, some aspects need more explanation, especially for readers who are not familiar with the topic/ described methods. Also, the added value of the proposed method, compared to existing methods, should be made clearer.
Abstract
- It is not common to use references in the abstract. Also, this is sentence is very similar to the first sentence of the introduction.
- Qualitative PAR method à explanation is needed or use the whole definition instead of the abbreviation (i.e. PAR). Also, this comment refers to “STEM learning opportunities” and “IRB review”.
- What is meant by the sentence: “..aspect of the lived experience of place, existing behavior..”
- The main advantage mentioned in the abstract is that “MBM does not require strict data collection protocols or statistical analysis”. However, are statistical analyses a bad thing? It should be made clearer why this method is better/ more suitable in this type of research compared to the quantitative method.
- These key words are very similar: behaviour mapping; behavioural mapping. I would suggest to use “placemaking” as a key word.
Introduction
- “By exploring aspects of the lived experience, participants understand how to transform that experience (6).” What is meant by transform the experience?
- References are needed to explain the definition of “place” and “space”, this is now missing
- More explanation, based on existing literature, is needed why existing methods are not effective for facilitating PAR. Also, a description of these existing methods is needed.
- The main aim or research question should be made more explicitly in the introduction
Material and methods
- It is mentioned that the quantitative behavioral mapping needs intensive data collection efforts. What is the difference with the proposed method, which also needs a lot of effort? This could be explained more extensively.
Results
- “Successful efforts occur simultaneously from bottom-up and top-down efforts” à how does this relate to your methodology?
Conclusion
- Future research is missing
- Is the methodology tested, using a case study?
- What are limitations of this method?
Author Response
Point 1: It is not common to use references in the abstract. Also, this is sentence is very similar to the first sentence of the introduction.
Response 1: I removed the quote in order to remove the reference which also remedied the similarity issue.
Point 2: Qualitative PAR method à explanation is needed or use the whole definition instead of the abbreviation (i.e. PAR). Also, this comment refers to “STEM learning opportunities” and “IRB review”.
Response 2: With the word limitation in the abstract, I refrained from using full terms and opted for abbreviations. In the text the full term is used at the first mention of PAR and STEM. Subsequent usage relies on the abbreviation which is typical. I did neglect to use the full term 'Institutional Review Board' at the first mention of IRB which I corrected. In reviewing, IRB is the US standard for academic work. Therefore, I explained the function of IRB, so international readers can make connections to similar organizations in their countries.
Point 3: What is meant by the sentence: “..aspect of the lived experience of place, existing behavior..”
Response 3: I expanded on this in the introduction.
Point 4: The main advantage mentioned in the abstract is that “MBM does not require strict data collection protocols or statistical analysis”. However, are statistical analyses a bad thing? It should be made clearer why this method is better/ more suitable in this type of research compared to the quantitative method.
Response 4: Statistical analysis is very good, of course, but may be out of reach for youth. The shift from quantitative to qualitative makes the method more youth-friendly.
Point 5: These key words are very similar: behaviour mapping; behavioural mapping. I would suggest to use “placemaking” as a key word.
Response 5: Placemaking appears in the title and therefore would appear in a search. I don't duplicate words that appear in the title as keywords. I also wanted to account for the US and British spelling of 'behavior' in order to make my article more searchable for an international audience.
Point 6: “By exploring aspects of the lived experience, participants understand how to transform that experience (6).” What is meant by transform the experience?
Response 6: The lived experience is captured in the mundane activities of daily life. Behavior would be included in these mundane activities. I expanded on this in the article.
Point 7: References are needed to explain the definition of “place” and “space”, this is now missing
Response 7: I added a reference.
Point 8: More explanation, based on existing literature, is needed why existing methods are not effective for facilitating PAR. Also, a description of these existing methods is needed.
Response 8: I expanded my argument for creating MBM as a PAR method. The argument doesn't assume that existing methods are inadequate. The argument proposes that MBM could be another tool in a researcher's toolbox. A description of existing methods is beyond the scope of the article. I identified methods that other researchers use for placemaking PAR.
Point 9: The main aim or research question should be made more explicitly in the introduction
Response 9: Reviewer 3 suggested reorganizing the article which highlighted the main aim in the introduction. The last paragraph of the introduction explains the aim of the article.
Point 10: It is mentioned that the quantitative behavioral mapping needs intensive data collection efforts. What is the difference with the proposed method, which also needs a lot of effort? This could be explained more extensively.
Response 10: Behavior mapping requires more data in order to perform statistical analysis which is not the case for MBM. I explained this in the revision.
Point 11: “Successful efforts occur simultaneously from bottom-up and top-down efforts” à how does this relate to your methodology?
Response 11: I explained this in the limitations.
Point 12: Is the methodology tested, using a case study?
Response 12: No. The methodology is created from my experience--years of working with children in placemaking PAR projects and my interest in behavior mapping.
Point 13: What are limitations of this method?
Response 13: I added limitations in the conclusion.
Reviewer 2 Report
This article suffers from a lack of clarity in the research. The tone of the article is non-academic or scientific as the language seems a set of instruction. The result is unknown, and there are no figures to explain the placemaking process. The relevance of the youth and their placemaking is not written.
Overall, this article needs to be revised to explain the quanititative and qualitative methods and the outcome or findings fully.
Occasionally there is a grammatical and structural issue in writing.
Author Response
Point 1: This article suffers from a lack of clarity in the research.
Response 1: I viewed this point as a general/opening statement. The reviewer goes on to identify areas that need clarification in their comments which I address.
Point 2: The tone of the article is non-academic or scientific as the language seems a set of instruction.
Response 2: I chose the open access route in order to grant practitioners free access to the article. Since the audience is both academic and practitioners, I think the tone is appropriate. The article introduces a new method and requires a set of instructions.
Point 3: The result is unknown, and there are no figures to explain the placemaking process.
Response 3: I revised the results section significantly.
Point 4: The relevance of the youth and their placemaking is not written.
Response 4: I mentioned this in the introduction and conclusion. Based on the comment, I didn't explain adequately. Therefore, I expanded on the relevance in both the introductions and conclusions.
Point 5: Overall, this article needs to be revised to explain the quantitative and qualitative methods and the outcome or findings fully.
Response 5: I didn't want to focus on the quantitative method in as much detail as the qualitative method since there are numerous sources explaining the quantitative method. The qualitative method is proposed and required a step by step explanation. I expanded on the findings of the qualitative method.
Point 6: Occasionally there is a grammatical and structural issue in writing.
Response 6: I reviewed and corrected.
Reviewer 3 Report
I want to thank you for the opportunity to review this manuscript. The time spent creating and shipping it is greatly appreciated. The document suggests adapting a quantitative research method into a qualitative one. However, its presentation, theoretically poorly bases, and, in my opinion, disorganized, limits the contributions of the work to the field of research.
In this way, I consider that for its possible publication in the magazine, it would be necessary to previously carry out an in-depth analysis of behavioral mapping, justifying the need to go further and develop MBM. Currently, the introduction is scarce and the motivation that leads to the development of this work is not sufficiently supported.
Likewise, the presentation of the work would improve significantly if the sections were grouped correctly. For example, the method presents the origin of behavioral mapping that would best fit in the introduction section. In addition, an applied case is presented in different lines and paragraphs, which would be beneficial to present fully in the results section. In this way, the structure of the work would improve significantly, making it easier to read and follow the argument of the manuscript.
One suggestion I make to the author is to include a comparative table between the two methods, illustratively, in the method section.
In short, in my opinion, the restructuring and organization of the manuscript, together with the expansion of the theoretical justification and justification, would significantly improve the work.
Thank you for your contribution.
Author Response
Point 1: In this way, I consider that for its possible publication in the magazine, it would be necessary to previously carry out an in- depth analysis of behavioral mapping, justifying the need to go further and develop MBM.
Response 1: An in-depth analysis of behavior mapping is beyond the scope of this article. There are other articles which do this and are referenced in the paper. I expanded the justification for MBM in the introduction.
Point 2: Currently, the introduction is scarce and the motivation that leads to the development of this work is not sufficiently supported.
Response 2: I expanded the introduction to include the motivation in developing MBM.
Point 3: Likewise, the presentation of the work would improve significantly if the sections were grouped correctly. For example, the method presents the origin of behavioral mapping that would best fit in the introduction section.
Response 3: I moved the language about the quantitative method to the introduction.
Point 4: In addition, an applied case is presented in different lines and paragraphs, which would be beneficial to present fully in the results section.
Response 4: The creation of the method is based on years of experience facilitating placemaking projects with youth. I used an example to illustrate how MBM results could be used. I eliminated the example because its presence confused the issue and reworked the results section.
Point 5: One suggestion I make to the author is to include a comparative table between the two methods, illustratively, in the method section.
Response 5: I definitely see value in presenting information graphically; however, I don't think a table is necessary. Throughout the entire article I compare the two methods. There was no way to condense the comparison in order to include the table. Reorganizing the article made the comparison clearer.
Round 2
Reviewer 1 Report
Thank you for the revision. The paper is well improved and the added value of the proposed method is made clearer. Also, the mehodology is explained in more detail. However, I have still some comments (see below). The main comment is related to the results section, which describes the focusgroup methodology. No results are presented, so it is not possible to understand whether the proposed method is valid.
- Line 11:..., namely modified behavior mapping (MBM)
- Line 20: which makes
- Line 21: children or do you mean youth?
- Line 23: in order to create design programs or designs of place --> this remains a bit vaque. For example, this method could be usefull for urban planners to design places that support desired behavior.
- It should also be made clearer in the paper why youth are important in the design process of urban places (?).
- Line 36: while the focus of placemaking activities is empowering people to imagine and create places, no particular focus occurs on the inclusive selection of the people involved. --> references are missing? and focus of whom? from urban planners/ academics/ government?
- Line 50: More and more attention is focused on empowering youth to be agents of change in placemaking --> by whom? why are youth agents of change? This could be elaborated more in the paper. Also, a reference for this is missing.
- Line 63: Many research methods fall within the purview of placemaking PAR with youth. --> name the references when you mention this.
- Line 64: the Welch et al., --> reference number is missing
- Also, it is not clear what you mention by youth? This should be explained (i.e. which age group). Some previous studies focus on children, so what is the difference? For example, Derr et al. (2018).
- Do you mean by place and urban public space or any indoor/outdoor place?
- Line 103/104: I think the following snetences are somewhat the same:In the article, a modification of behavior mapping is proposed in order to assist youth in placemaking PAR. Modified behavior mapping (MBM) transforms the quantitative method into a proposed PAR method to assist youth in placemaking.
- Line 129: child friendly or youth friendly?
- The results section is a bit strange, a focusgroup is also the methodology? Did you tested the method on youth in a case study? Did you found any results? I think the structure of this part could be improved. See for example, https://www.mdpi.com/1660-4601/17/14/5205/htm.
- The practical implications of the method should be stated more clearly in the conclusion section.
Author Response
Point 1: Line 11:..., namely modified behavior mapping (MBM)
Response 1: I added 'namely'.
Point 2: Line 20: which makes
Response 2: I changed 'make' to 'makes'.
Point 3: Line 21: children or do you mean youth?
Response 3: I changed children to youth.
Point 4: Line 23: in order to create design programs or designs of place --> this remains a bit vaque. For example, this method could be usefull for urban planners to design places that support desired behavior.
It should also be made clearer in the paper why youth are important in the design process of urban places (?).
Response 4: I omitted "design program and designs of place" in the abstract since the word limit prevents explanation and followed the reviewers suggestion of focusing on who would benefit from the method.
I explained the importance of youth in the design process in line 51-52 and added language at the end of the results section.
Point 5: Line 36: while the focus of placemaking activities is empowering people to imagine and create places, no particular focus occurs on the inclusive selection of the people involved. --> references are missing? and focus of whom? from urban planners/ academics/ government?
Response 5: These are offered as introductory sentences connecting placemaking and PAR. I'm making the point that inclusion is not a mandatory component of placemaking as it is in PAR. By combining the two approaches, vulnerable populations are empowered to transform their environments.
Point 6: Line 50: More and more attention is focused on empowering youth to be agents of change in placemaking --> by whom? why are youth agents of change? This could be elaborated more in the paper. Also, a reference for this is missing.
Response 6: This is just an introductory sentence. Within the paragraph, I provide details about whom and references.
Point 7: Line 63: Many research methods fall within the purview of placemaking PAR with youth. --> name the references when you mention this.
Response 7: This is just an introductory sentence. The references appear within the paragraph.
Point 8: Line 64: the Welch et al., --> reference number is missing
Response 8: I add the reference numbers.
Point 9: Also, it is not clear what you mention by youth? This should be explained (i.e. which age group). Some previous studies focus on children, so what is the difference? For example, Derr et al. (2018).
Response 9: I added the ages of the children reported in the 5 references within that chapter.
Point 10: Do you mean by place and urban public space or any indoor/outdoor place?
Response 10: In lines 30-35, I explain 'place'. I decided to define 'place' in terms of place versus space in order to make the link to the lived experience and connect to PAR.
Point 11: Line 103/104: I think the following snetences are somewhat the same:In the article, a modification of behavior mapping is proposed in order to assist youth in placemaking PAR. Modified behavior mapping (MBM) transforms the quantitative method into a proposed PAR method to assist youth in placemaking.
Response 11: I omitted "to assist youth in placemaking" in the second sentence.
Point 12: Line 129: child friendly or youth friendly?
Response 12: I changed to youth.
Point 13: The results section is a bit strange, a focusgroup is also the methodology? Did you tested the method on youth in a case study? Did you found any results? I think the structure of this part could be improved. See for example, https://www.mdpi.com/1660-4601/17/14/5205/htm.
Response 13: The method is based on years of experience working with the quantitative method behavior mapping and working with youth in placemaking exercises. This method evolved within the context of placemaking exercises.
I introduced the framework for MBM at the end of the first paragraph in the Results section in order to better frame the section.
I also added a paragraph at the end of the results section to address your earlier comment in point 4 about why youth inclusion in placemaking is important.
Point 14: The practical implications of the method should be stated more clearly in the conclusion section.
Response 14: I added more language about building youth agency and inclusion.
Reviewer 2 Report
I have checked it and I can confirm that the author has responded sufficiently. The paper can be approved for publication.Reviewer 3 Report
I believe that most of the recommendations have been carried out. Thanks for your contribution.
Author Response
Point 1: Does the introduction provide sufficient background and include all relevant references? Can be improved.
Response 1: I utilized the comments of Reviewer 1 to improve.
Point 2: Are the methods adequately described? Can be improved.
Response 2: Reviewer 1 commented that the methods section was clearer and much improved from the first version. Without clear direction, I did not change this section.
Point 3: Are the results clearly presented? Can be improved.
Response 3: I utilized the comments of Reviewer 1 to improve.